# Prevalence and Risk Factors Associated with Malaria among Children Aged Six Months to 14 Years Old in Rwanda: Evidence from 2017 Rwanda Malaria Indicator Survey

**DOI:** 10.3390/ijerph17217975

**Published:** 2020-10-30

**Authors:** Faustin Habyarimana, Shaun Ramroop

**Affiliations:** School of Mathematics, Statistics and Computer Sciences, Pietermaritzburg Campus, University of KwaZulu-Natal, Private Bag X01, Scottsville 3209, South Africa; RamroopS@ukzn.ac.za

**Keywords:** children, malaria, RDT, 2017 RMIS, Rwanda, survey logistic regression

## Abstract

Malaria is a major public health risk in Rwanda where children and pregnant women are most vulnerable. This infectious disease remains the main cause of morbidity and mortality among children in Rwanda. The main objectives of this study were to assess the prevalence of malaria among children aged six months to 14 years old in Rwanda and to identify the factors associated with malaria in this age group. This study used data from the 2017 Rwanda Malaria Indicator Survey. Due to the complex design used in sampling, a survey logistic regression model was used to fit the data and the outcome variable was the presence or absence of malaria. This study considered 8209 children in the analysis and the prevalence of malaria was 14.0%. This rate was higher among children aged 5–9 years old (15.6%), compared to other age groups. Evidently, the prevalence of malaria was also higher among children from poor families (19.4%) compared to children from the richest families (4.3%). The prevalence of malaria was higher among children from rural households (16.2%) compared to children from urban households (3.4%). The results revealed that other significant factors associated with malaria were: the gender of the child, the number of household members, whether the household had mosquito bed nets for sleeping, whether the dwelling had undergone indoor residual spraying in the 12 months prior to the survey, the location of the household’s source of drinking water, the main wall materials of the dwelling, and the age of the head of the household. The prevalence of malaria was also high among children living in houses with walls built from poorly suited materials; this suggests the need for intervention in construction materials. Further, it was found that the Eastern Province also needs special consideration in malaria control due to the higher prevalence of the disease among its residents, compared to those in other provinces.

## 1. Introduction

Malaria is one of the main public health problems in the world and is an important cause of morbidity and mortality among children in Sub-Saharan Africa. Worldwide, malaria incidence has significantly decreased between 2010 and 2017, from 71 to 57 cases per 1000 of the population at risk [1]. In 2018, worldwide, there were approximately 405,000 deaths from malaria, a decrease from 416,000 deaths in 2017. Children were more vulnerable compared to other age groups where they accounted for 67% of all global malaria deaths (272,000) in 2018 [1]. The infectious disease is most prevalent in Sub-Saharan Africa and in India, as 19 countries from Sub-Saharan Africa and India disproportionately carried 85% of the global malaria burden [1].

In east African countries, the prevalence of malaria in children under five years has decreased in general, but it is still high. For instance, the prevalence of malaria from the last two recent studies in Burundi was 22% in 2013 and 27% in 2016–2017; in Kenya it was 5% in 2015; in Rwanda it was 2% in 2015 and 7% in 2017; in Tanzania it was 14% in 2016 and 7% in 2017; in South Sudan it was 32% in 2017 and in Uganda it was 30% in 2015 and 30.3% in 2016 [2,3,4,5,6,7].

In Rwanda, malaria is also a public health concern because the entire population is at risk of contracting this infectious disease. Malaria has been the main cause of morbidity and mortality in Rwanda for several years [2]. It is a life-threatening disease caused by infection through the bite of a female *Anopheles* mosquito. It is most prevalent among children and among pregnant women in malaria-prone areas such as the Eastern Province and Southern Province which together accounted for 79% of malaria cases in Rwanda in 2017 [8].The Government of Rwanda has established a mandatory health insurance scheme known as *mutuelle de santé* which has helped the citizens of Rwanda to access private or public health centers, clinics and hospital services. In addition, the government has implemented various strategies to prevent malaria, such as distributing insecticide-treated mosquito ets (ITNs) and employing indoor residual spraying (IRS) within the households. As a result, the prevalence of malaria among the population of Rwanda dropped drastically up to 2011 [2]; for instance the prevalence of malaria among women of child-bearing age was 2.6% in 2007–2008 and dropped to 0.7% in 2010; in 2015 it was 0.6%, and in 2017 it rose to 5% [9]. The prevalence of malaria among children under five years was 2.6% in 2007–2008 and dropped to 1.4% in 2010 [9].

Despite the above-mentioned strategies, the prevalence of malaria among children under five years old in Rwanda has increased by 6% from 2010 to 2017 [9,10]. Further, the results from [9] showed that the prevalence of malaria among children aged five years to 14 years was higher compared to other age groups (15.1%) and it was followed by the prevalence of malaria among children aged six months to 59 months (11.8%); where the overall prevalence of malaria in the Rwandan population was 7% [9]. Consequently, these two age groups need special consideration. Therefore, the main objectives of this study were to identify the factors associated with malaria among children aged six months to 14 years and to assess the prevalence of malaria among this age group in Rwanda. To the best of our knowledge, no other study has been carried out in Rwanda with these objectives.

## 2. Materials and Methods

### 2.1. Source of Data

This study used cross-sectional secondary data from the 2017 Rwanda Malaria Indicator Survey (RMIS). This survey was conducted from October to December 2017, at the peak of malaria season. The survey used the sampling frame from the 2012 Rwanda Population and Housing Census [11]. The data were collected using a two-stage stratified sampling method. In the first stage, a total of 170 primary sampling units were selected. The sampling was done using probabilities proportional to the number of households in the village. In the second stage, using a systematic sampling technique, 30 households were selected from each primary sampling unit. In total, the sample size was 5100 households. All women aged 15–49 years who were either permanent residents of or visitors to the selected households were eligible for the interview. The malaria testing was done after obtaining consent from adults aged 15 years or older and from the parents or guardians in the case of children aged six months to 14 years [9].

#### 2.1.1. Dependent Variable

The response variable of interest in this study is the malaria rapid diagnostic test (RDT) result. The RDT is immune-chromatography form of test which assists in diagnosing malaria by detecting specific antigens (proteins) produced by malaria parasites in the blood of infected person. Some RDT can detect only one species (plasmodium falciparum) while others detect multiple species. If malaria antigens are present, then the person tests positive. If malaria antigens are not present, the person tests negative [12]. Therefore, the response variable was binary, 1 indicating the presence of malaria infection and 0 indicating no malaria infection. A total of *n* = 8209 children aged six months to 14 years old were considered in this study.

#### 2.1.2. Independent Variable

The potential explanatory variables considered in this study were demographic and socio-economic factors and factors such as age of the head of the household (continuous), age of the child (six to 59 months, five to nine years, 10 to 14 years), sex of the child (female, male), number of household members (or size of the family); socio-economic factors such as wealth index of the household (poorest, poor, middle, rich, richest), ownership of radio (yes, no), ownership of television (yes, no), use of electricity (yes, no), wall material of dwelling (uncovered, covered adobe, cement, others), roof material of dwelling (metal, ceramic tiles, others), floor material of dwelling (earth/sand, cement/ceramic tiles, others), number of living rooms (1,2,3 and more); environmental and sanitation factors such as use of mosquito net for sleeping (yes, no), whether the dwelling was sprayed for mosquitoes in the twelve months prior to the survey (yes, no, do not know), place of residence (urban, rural), province of residence (Kigali, East, South, West, North) and location of drinking water source (dwelling/yard, other). The justification as to the choice of these variables from a rigorous theoretical framework was that these variables have been found to be statistically significant in the existing literature [12,13,14,15] to name a few.

### 2.2. Statistical Methods

#### Statistical Analysis

The surveys, were done were based on multi-stage sampling, stratified and cluster sampling with unequal probability of selection for elements to be included in the survey, known as complex survey design. When modelling the data collected from these surveys, the complexity of the sampling design must be taken into consideration. Therefore, in order to account for the effect of stratification, clustering, sampling weights and to relax the assumption of independence of observation of the ordinary logistic regression model, the present study used a survey logistic regression model for the data analysis. Failure to account for clustering and sampling weights may lead to underestimation of the variability and, consequently, to wrong inference. In general, the theory of ordinary logistic regression and survey logistic regression is the same; they differ only in variance estimation. They are both members of the generalized linear models, and a maximum likelihood is used to estimate the parameters. The model formulation used in this study is discussed in detail as follows:

Let $y_{ijk}$ denote the malaria status of child *i* from stratum *j* and cluster k, with i=1, 2, 3, …, 8209, j=1, 2, 3, …, 60 and k=1, 2, 3, …, 170. The outcome variable is defined as a dichotomous variable such that $y_{ijk}=1$ if the child *i* has malaria and $y_{ijk}=0$ otherwise.

The current study assumes that the outcome variable $y_{ijk}$ is Bernoulli distributed as $\;y_{ijk}|\mu_{ijk}∼Bernoulli\;\left(\mu_{ijk}\right)$, where $\mu_{ijk}$ is known as mean and it is given by $\;E\left(y_{ijk}\right)=\mu_{ijk}$, and it is related to the covariates as follows:
$$g\left(\mu_{ijk}\right)={x^{\prime }}_{ijk}\beta$$
where g(.) is the logit link function, β is a vector of unknown model parameters.

The analysis in this study was done using SAS Proc Surveylogistic from SAS software version 9.4 (SAS Institute, Cary, NC, USA). The Taylor series method was used as a variance estimator. The model fit statistics were done based on Akaike information criteria (AIC) and −2 Log-Likelihood (−2LogL) principles. The model test was done based on the likelihood ratio, score and the Wald test principles.

## 3. Results and Interpretations

The current study considered 8209 children aged six months to 14 years; 4053 of them were female, whereas 4156 of them were male. Of the sample, 14.0% tested positive for malaria infection and 86.0% tested negative. It is observed from Table 1, that 12.4% of participants were from Kigali, 20.8% from the Southern Province, 22.9% from the Western Province, 18.5% from the Northern Province and 25.5% from the Eastern Province. It is observed from the table that 5.0% of dwelling wall materials were cement, 29.9% covered adobe, 32.2% uncovered adobe and 32.9% other materials. A total 42.2% of households had a radio whereas 57.8% did not have a radio. It is also observed from the same table that most of the respondents, 82.4%, were from rural areas and 17.6% were from urban areas. Table 1 shows that 10.8% of the respondents had a television and 89.2% did not have a television. It is observed from Table 1 that 2.0% of the households had water in their dwelling/yard and the majority of the respondents, 98%, had to source drinking water from locations other than their dwelling/yard. Table 1 shows that 93.8% of participants had heard of malaria, whereas 6.2% of them had not heard about malaria. It is observed that the median age of the head of the household was 40 years, the minimum age was 16 years and the maximum age was 95 years.

The association between RDT results and various potential factors was tested by the chi-square statistical test at a 5% level of significance and the results are summarized in Table 2. Any variable that was statistically significant in the cross-tabulation was included in the final analysis and we have only reported the variables that are significant at the 5% level of significance. It was observed from this result that the prevalence of malaria among children was slightly higher among females: 14.2% compared to males at 13.8% (Table 2).

The prevalence of malaria among children was highest in the Eastern Province, 31.1% compared to other provinces and it was lowest in the Northern Province at 2.0%. It was observed that the prevalence of malaria detected by RDT was higher among children from rural residential areas—16.2%—than their urban counterparts—3.4%. The results from the cross-tabulation showed that the prevalence of malaria among children was highest among children from the poorest households (19.4%) and lowest among children from the richest families (4.3%). The prevalence of malaria was lowest among children under five years (11.7%) and highest among children aged five to nine years old (15.6%) (Table 2).

The study found a significant association between the wall materials of the dwelling and results of RDT (*p*-value < 0.001) (Table 2). The prevalence was lowest among children from households where the main wall materials were cement (5.4%) and it was highest among children from households with wall materials other than cement, for example, covered adobe and uncovered (21.4%). The prevalence of malaria was 0.6% among children from households where they had water inside the dwelling or yard and 14.2% among the children from households where the source of water was elsewhere. The prevalence of malaria was higher among children from households which did not have electricity (17.1%) and lower among children from households which had electricity (7.6%) (Table 2).

The current study considered the individuals’ available media and it was found that the prevalence of positive malaria RDT outcomes was lessened by the ownership of media such as a radio and a television. The prevalence of malaria was 10.7% among the children from households that owned a radio and 16.4% among the children from households that did not have a radio. Children from households that had a television were less associated with the prevalence of malaria (4.4%) and the prevalence of malaria was 15.1% among children from families which did not have a television.

The model fit is presented in Table 3, where Akaike information criterion (AIC), the Schwarz criterion (SC) and the negative of twice the log likelihood (−2LogL) were tested. AIC, SC and −2LogL are used to compare the reduced model (model of intercept only) and the full model (model of intercept and covariates). It is observed from Table 3, that AIC, SC and −2LogL were smaller for the full model (model with intercept and covariates) compared to the reduced model (model with intercept only); this means that the full model was the better model fit.

The global null hypothesis states that all parameters are equal to zero. This was tested, with the results presented in Table 4, where the likelihood ratio, score and Wald tests were all highly significant (*p*-value < 0.0001), which means that at least one of the parameters is not equal to zero.

The results from multiple survey logistic regression are summarized in Table 5.

In order to avoid possible confounding effects, the two-way interaction between potential variables was considered, but none were found to be significant. The results from multiple survey logistic regression are summarized in Table 5. The findings of this study revealed that the age of the child, use of mosquito bed nets for sleeping, province of residence, the wealth quintile of the household, place of residence, location of (drinking) water source, the number of the household members, age of the household head, number of living rooms, type of wall materials, type of roof material, and whether the dwelling had been sprayed against mosquitoes in the last 12 months were found as risk factors associated with malaria infection among children aged six months to 14 years in Rwanda.

The place of residence was significantly associated with a risk of malaria in children aged six months to 14 years in Rwanda (*p*-value = 0.0175). It was found that the probability of having malaria was higher among children living in rural areas than among their urban counterparts (Adjusted Odds ratio (OR) = 2.792, CI = 1.199, 6.498). The present study revealed that the type of wall materials was significantly associated with the risk of malaria among children aged six months to 14 years. It was observed from the results that children living in houses with re-used wood as the main wall material were 2.234 times more likely to have malaria than children living in houses with wall materials other than bamboo with mud or uncovered and a covered adobe. However, this study did not find a significant difference between children living in houses of bamboo with mud or an uncovered adobe and children living in houses with main wall materials other than reused wood.

It was observed from Table 4 that the likelihood of contracting malaria infection among children aged six months to 14 years in Rwanda reduced with increasing family wealth quintile. The children from the poorest, poor and rich households were found to be more vulnerable to malaria than children from the richest households. It was observed that children from the poorest, poor or rich households were, respectively, 2.903, 2.381 and 1.705 times more likely to have malaria than children from the richest households. The present study did not find a significant statistical effect between children from middle income households and children from the richest households.

The number of household members was also found to be a significant risk factor associated with malaria among children in Rwanda (Table 5). The results revealed that for a unit increase in family size, the number of infected children in the family increased by 8.8% (OR: 1.088, *p*-value = 0.0280). The location of source of drinking water was also found to be a significant factor associated with malaria among children aged six months to 14 years (*p*-value = 0.0213). The results showed that children from a household where they used water from their dwelling or yard were less likely to have malaria, as compared to children from a household where they used water from locations other than the dwelling or yard. The risk of malaria was 0.106 times less among children from a family where they used water from their dwelling compared to children from households where they used water from other locations. The age of the head of the household was also found to be a significant risk factor associated with malaria among children aged six months to 14 years (*p*-value = 0.0415). The results revealed that one unit increase in age of the head of the household increased the odds of a positive malaria test by 0.9% (OR: 1.009, *p*-value = 0.0415) among children of that household.

The province of residence was also found to be a significant risk factor for malaria. The risk of having malaria was 2.490 times more likely among children from the Eastern Province compared to children from Kigali City. It was also found that children from the Northern Province and from the Western Province were, respectively, 0.097 and 0.145 times less likely to have malaria compared to children from Kigali City. The present study did not find a significant difference among children from the Southern Province and children from Kigali City, but the results showed a positive association. The number of living rooms was also found to be a significant risk factor associated with malaria prevalence among children aged six months to 14 years in Rwanda. Children from a household with two living rooms were 0.7 (OR = 0.723, CI = 0.489, *p*-value = 0.0325) times less likely to have malaria than children from a household with one living room. The current study did not find a significant difference between children from a household with three or more living rooms and children from a household with one living room.

The age group of the children was found to be significantly associated with factors of malaria among children (*p*-value = 0.0065). Children from the age group 5–9 years were 1.357 times more likely to have malaria than those from age group six months to 59 months. However, the present study did not find a significant difference between children aged less than five years and those aged from 10 to 14 years.

It was observed from Table 5 that the use of mosquito nets for sleeping was significantly associated with the risk factor of malaria (*p*-value = 0.0349). The results from this study showed that the risk of malaria among children lessened with the use of mosquito bed nets for sleeping. Children from households where mosquito nets were not used for sleeping were 1.5 times more susceptible to malaria infection (OR = 1.499) compared to their counterparts from households where they used mosquito nets for sleeping. It was also observed that IRS was significantly associated with the risk factors of malaria (*p*-value = 0.0041). Children from households that had not been sprayed against mosquitoes in the 12 months prior to the survey were 2.266 times more likely to have malaria than those from households which had been sprayed against mosquitoes in the same time period.

## 4. Discussion

Malaria is still the main cause of morbidity and mortality among children in Sub-Saharan African countries. The knowledge of the risk factors associated with malaria among children aged six months to 14 years provides an insight into the methods and policies that can be used to combat this public health problem effectively.

The government of Rwanda has invested great efforts to improve healthcare programs, particularly at the level of primary healthcare. It has implemented several strategies for monitoring and evaluating malaria control on a regular basis, where the focus is mainly to reduce malaria morbidity and mortality.

The present study was carried out based mainly on data from the 2017 Rwanda Malaria Indicator Survey. The main objective of this study was to identify the factors associated with malaria status among children aged six months to 14 years and to determine the prevalence of malaria among the children from this age group. In the current scientific setting we have used the gender of the child, the age of the child, the age of the head of the household, the number of household members, the number of living rooms, wall material of dwelling, roof material of dwelling, floor material of dwelling, province of residence, place of residence (rural or urban), use of ITNs for sleeping, location of source of drinking water.

It was found that the likelihood of malaria increases with the increasing size of the household. This is in line with the findings of other studies such as [13,14]. The reason for this is that, if they are many individuals in the household, when one of them is infected, he/she could serve as reservoir for others [15,16].

It was found in this study that exposure to media such as a radio and a TV was not significant. The government of Rwanda sometimes communicates messages regarding malaria to the population through *Umuganda* (compulsory collective work done on the last Saturday of every month), which happens at the same time countrywide in all grass root administrations, or through *Abajyanama Bubuzima* (community health workers) and this may be the reason why the media does not have an effect. This is confirmed by the high rate of malaria awareness in women aged 15 to 49 years old (81%) [9]. In this study we found that malaria prevalence was higher among children from rural areas than children from urban settlements. This was also found in various similar studies [17,18,19,20,21,22]. The cause may be that people living in rural settlements lack sufficient access to health-care facilities and have most of the cases of poor housing conditions that expose children to malaria transmitting vectors. Moreover, children of school-going age from rural areas may get infected while travelling to and from school as some pass through forests or agricultural plantations where malaria vectors are abundant; they may thus be more susceptible to mosquito bites than their urban peers.

The use of mosquito bed nets was found to play an essential role in preventing malaria infection among children aged six months to 14 years in Rwanda. The present study found that children from the families where mosquito bed nets were not used when sleeping were more vulnerable to malaria infection than children from families where they were used. While this is consistent with other findings from similar studies [13,23,24,25], the result conflicts with the findings of [14,17].

The current study found that the malaria infection rate among children increases with their increasing age. A similar finding has been observed in previous studies [13,14,17,26,27,28,29]. This may be since the use of mosquito nets for sleeping in Sub-Saharan African countries generally reduces with increasing age among children. In Rwanda, the 2017 RMIS showed that 70.0% of children under five years old, 58.9% of children between five and 14 years old, 63.8% of people aged 15 to 34 years old and 70.3% of people aged 50 years and older, slept under any type of mosquito net [9].

The results from this study revealed a higher infection rate of malaria among children from poor families compared to children from the richest families. This finding is consistent with previous studies [14,19,20,30,31,32]. The poorest households may have had poor quality houses that created an environment conducive to the spread of malaria [33]. Previous studies from Africa have found that better constructed houses are associated with a lower risk of malaria [13,34]. These findings show that effective malaria control and malaria eradication are not isolated efforts of a malaria program only, but they work in unison with the effort of improving the socio-economic status of the population [35].

The results of this study showed a significant association between the type of house wall materials and malaria prevalence. This finding is consistent with similar studies [13,36]. It was found that children from families living in poorly constructed housing, with particular wall materials had a higher chance of contracting malaria and this is consistent with [36].

The province of residence was shown to be significantly associated with the risk of malaria, where children from the Northern Province and from Western Province had a low risk of contracting malaria compared to children from Kigali City. This finding is consistent with that of [9]. This is understandable because the Northern Province is made up mainly of mountains with relatively high altitude; it is a volcanic region with cold weather, known to be unfavorable for mosquitoes [37,38]. The results from this study also revealed that the malaria prevalence was higher among children from the Eastern Province compared to children from the other provinces. This finding is consistent with the findings from other studies [2,39]. This may be due to the Eastern Province’s generally low altitude, compared to other Rwandan provinces and its many open water sources, which serve as reservoirs for mosquito breeding sites and thereby increase the risk of malaria infection [37,38,40].

## 5. Conclusions

In the present study, survey logistic regression was used to account for the complexity of the design which led to appropriate statistical inference with accuracy. The findings from this study have provided insights into the demographic, socio-economic and environmental factors such as age of the household head, size of the household, wealth index of the household, whether the household was sprayed for mosquitoes in the 12 months prior to the survey, use of mosquito bed nets for sleeping, the dwellings’ main wall materials, location of source of water for drinking, place of residence, province of residence, and age of the children, which were significant factors associated with malaria prevalence among children aged six months to 14 years. This study found that the use of IRS and insecticide treated nets is an important intervention to prevent malaria infection among children aged six months to 14 years. It was found that the provinces had different risk levels, with the Eastern Province being highest risk. This suggests special consideration should be given for malaria control, such as the provision of ITNs, in this province. Therefore, the findings from this study may help public health planners, policy-makers and other related institutions in effective decision-making related to malaria control, especially as it pertains to children aged six months to 14 years. The higher prevalence of malaria was among children from low income households, which suggests the need for an intervention in households with lower income levels. The results from this study revealed a high prevalence of malaria among children aged five to nine years, which suggests the need for an intervention in this age group: for instance, increasing the use of mosquito nets among children of school-going age because it is in this age group that the use of mosquito nets for sleeping is low, compared to other age groups.

The limitation of this study is that the data are cross-sectional and more meaningful trends and patterns could be identified from a longitudinal analysis. Future research can possibly be undertaken through incorporation of spatial information.

## Figures and Tables

**Table 1 ijerph-17-07975-t001:** Descriptive statistics of the participants.

Variable	Category	% or M or Range
**RDT**	Positive	14.0
	Negative	86.0
**Province**	Kigali	12.4
	South	20.8
	West	22.9
	North	18.5
	East	25.5
**Wealth Index**	Poorest	21.0
	Poor	20.3
	Middle	20.0
	Rich	21.1
	Richest	17.6
**Age of Household Head (Continuous)**	Continuous	Median = 40, M = 95 (Min = 16)
**Main Wall Materials**	Uncovered	32.2
	Covered Adobe	29.9
	Cement	5.0
	Others	32.9
**Household has Radio**	Yes	42.2
	No	57.8
**Household has Electricity**	Yes	33
	No	67
**Household has Television**	Yes	10.8
	No	89.2
**Location of Water Source**	Dwelling/Yard	2.0
	Others	98.0
**Has Mosquito Net for Sleeping**	Yes	89.8
	No	10.2
**Dwelling has been Sprayed against Mosquitoes in the 12 Months Prior to Survey**	Yes	20.2
	No	79.6
	Do not Know	0.2
**Number of the Household Members**	Continuous	Median = 5, M = 13 (Min = 2)
**Has Heard of Malaria**	Yes	93.8
	No	6.2
**Floor Materials of Dwelling**	Earth/Sand	71.9
	Cement/Ceramic Tiles	4.2
	Others	23.9
**Roof Materials of Dwelling**	Metal	32.0
	Ceramic Tiles	65.5
	Others	2.5
**Sex of Children**	Female	32.0
	Male	68.0
**Ages of Children**	6–59 Months	34.3
	5–9 Years	33.6
	10–14 Years	32.1
**Place of Residence**	Rural	82.4
	Urban	17.6
**Number of Living Rooms**	1	45.7
	2	41.6
	3 and More	12.7

RDT: rapid diagnostic test.

**Table 2 ijerph-17-07975-t002:** Prevalence distribution of malaria among children aged six months to 14 years.

Malaria Status			
Variable	Negative N (%)	Positive N (%)	*p*-Value
**Province**			<0.0001
Kigali	939 (92.3)	78 (7.7)	
South	1387 (81.2)	321 (18.8)	
West	1813 (96.6)	64 (3.4)	
North	1484 (98.0)	31 (2.0)	
East	1440 (68.9)	651 (31.1)	
**Wealth index**			<0.0001
Poorest	1389 (80.6)	335 (19.4)	
Poor	1369 (82.2)	297 (17.8)	
Middle	1413 (86.1)	228 (13.9)	
Rich	1506 (87.0)	225 (13.0)	
Richest	1385 (95.7)	62 (4.3)	
**Age of Household Head (Continuous)**	Median = 40, Minimum = 16, Maximum = 95		
**Main Wall Materials of Dwelling**			<0.0001
Uncovered	2309 (87.5)	331 (12.5%)	
Covered Adobe	2240 (91.2)	215 (8.8%)	
Cement	389 (94.6)	22 (5.4%)	
Others	2125 (78.6)	578 (21.4%)	
**Household has Radio**			<0.0001
Yes	3095 (89.3)	371 (10.7)	
No	3967 (83.6)	776 (16.4)	
**Household has Electricity**			<0.0001
Yes	2502 (92.4)	205 (7.6)	
No	4560 (82.9)	942 (17.1)	
**Household has Television**			<0.0001
Yes	844 (95.6)	39 (4.4)	
No	6216 (84.9)	1107 (15.1)	
**Location of Source of Drinking Water**			<0.0001
Dwelling/Yard	159 (99.4)	1 (0.6)	
Others	6904 (85.8)	1145 (14.2)	
**Has Mosquito Net for Sleeping**			<0.0001
Yes	6421 (87.1)	947 (12.9)	
No	641 (76.2)	200 (23.8)	
**Dwelling has Been Sprayed against Mosquitoes in the 12 Months Prior to Survey**			<0.0001
Yes	1382 (83.2)	279 (16.8)	
No	5667 (86.7)	866 (13.3)	
Do not Know	14 (93.3)	1 (6.7)	
**Number of Household Members (Continuous)**	(Median = 5, Minimum = 2, Maximum = 13)		
**Has Heard of Malaria**			<0.0001
Yes	6591 (85.6)	1107 (14.4)	
No	472 (92.4)	39 (7.6)	
**Floor Materials of Dwelling**			<0.0001
Earth/Sand	4923 (83.4)	980 (16.6)	
Cement/Ceramic Tiles	1920 (93.2)	141 (6.8)	
Others	220 (89.8)	25 (10.2)	
**Roof materials of dwelling**			<0.001
Metal	4542 (84.5)	836 (15.5)	
Ceramic tiles	2347 (89.3)	281 (10.7)	
Others	174 (85.7)	29 (14.3)	
**Sex of children**			0.301
Female	3478 (85.8)	575 (14.2)	
Male	3584 (86.2)	572 (13.8)	
**Ages of children**			<0.0001
6–59 months	2407 (85.5)	407 (14.5)	
5–9 years	2331 (84.4)	431 (15.6)	
10–14 years	2324 (88.3)	309 (11.7)	
**Place of Residence**			<0.0001
Rural	5669 (83.8)	1097 (16.2)	
Urban	1394 (96.6)	49 (3.4)	
**Number of Living Rooms**			<0.0001
1	3275 (87.3)	477 (12.7)	
2	2942 (86.3)	469 (13.7)	
3 and more	846 (80.9)	200 (19.1)	

**Table 3 ijerph-17-07975-t003:** Model Fit Statistics.

Criterion	Intercept Only	Intercept and Covariate
AIC	6190.601	4946.108
SC	6197.486	5118.234
−2LogL	6188.601	4896.108

AIC: Akaike information criterion; SC: Schwarz criterion.

**Table 4 ijerph-17-07975-t004:** Testing global null hypothesis: Beta=0.

Test	F Value	Num DF *	*p*-Value
**Likelihood Ratio**	14.11	11.2615	<0.0001
**Wald**	7.12	24	<0.0001

* Num Df = Number of degrees of freedom.

**Table 5 ijerph-17-07975-t005:** Estimates and odds ratios of socio-economic, demographic and environmental factors on RDT.

Parameter	Estimate	StandardError	Pr > |t|	Adjusted Odds Ratio (OR)
**Intercept**	−4.9972	0.7013	<0.0001	0.007
**Province (Kigali = Ref)**				
East	0.9122	0.4394	0.0395	2.490
South	0.0127	0.4633	0.9782	1.013
West	−1.9323	0.5409	0.0005	0.145
North	−2.3294	0.7700	0.0029	0.097
**Place of Residence (Urban = Ref)**				
Rural	1.0037	0.4328	0.0217	2.728
**Number of Household Members**	0.0815	0.0387	0.0365	1.085
**Main wall materials of dwelling (Others = Ref)**				
Covered Adobe	0.2356	0.3689	0.5241	1.266
Uncovered Adobe	0.2822	0.4094	0.4915	1.326
Re-used Wood	0.8030	0.3722	0.0324	2.232
Bamboo with Mud	0.8425	0.4367	0.0555	2.322
Dirt	0.3786	0.3933	0.3372	1.460
**Has Mosquito net for Sleeping (Yes = Ref)**				
No	0.4049	0.1903	0.0349	1.499
**Dwelling has been sprayed against Mosquitoes in 12 Months Prior to Survey (Yes = Ref)**				
No	0.8178	0.2627	0.0022	2.266
Do not Know	0.0316	0.4866	0.9483	1.032
**Age of Household Head (Continuous)**	0.00863	0.00420	0.0415	1.009
**Number of Living Rooms**				
**(1 Room = Ref)**				
2	−0.3107	0.1773	0.0817	0.733
3	−0.3250	0.1507	0.0325	0.723
**Ages of Children (6–59 Months = Ref)**				
5–9 Years	0.3056	0.0953	0.0016	1.357
10–14 Years	0.1902	0.1110	0.0885	1.209
**Location of Source of Drinking Water (other than Dwelling/Yard = Ref.)**				
Dwelling/Yard	−2.2480	0.9665	0.0213	0.106
**Wealth index (Richest = Ref)**				
Poorest	1.0657	0.3403	0.0021	2.903
Poor	0.8675	0.3042	0.0049	2.381
Middle	0.5097	0.3000	0.0913	1.665
Rich	0.5316	0.2266	0.0202	1.702

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
