# Peer review of "Prevalence and Risk Factors Associated with Malaria among Children Aged Six Months to 14 Years Old in Rwanda: Evidence from 2017 Rwanda Malaria Indicator Survey"

_ijerph, 2020, doi:10.3390/ijerph17217975_

Round 1

Reviewer 1 Report

This is a very interesting paper and addresses a much needed epidemiological question of malaria risk factor in Africa which is usually known as a disease of poverty but needs to be addressed on regular bases.

the paper uses multiple logistic regression models to assess risk factors, despite the efforts it seems that the authors are not taking enough time to describe each and every element. below are my itemized comments.

Major comments:

1) The authors did not define their parameters as they start. They claim poor, poorer and poorest but provided no definition of what is defined as what? Also what makes a rural vs urban site rich or poor? as a person who is not from Ruwanda I was completely confused with this random use of the words without clarification

2) The model ignored completely the presence of windows and window nets. I assume all those houses have windows or open doors! including the cement ones. Is there a reason for ignoring that?

3) I am very confused with the age group description of the children, sometimes they are one sum, other times they are two groups, then on third occasions they are three groups, simplify and perhaps consolidate the groups. 

4) The paper completely avoided any descriptive graphs. there is such heavy text loaded with number and not enough graphs or even maps to describe the background! When you have four locations what makes it rural?

5) the authors completely ignored the biological questions. They state that women are more at risk of infection but never mentioned why! Is there an explanation? also what is the biology of malaria in Rwanda! while the paper is more of epidemiological nature it is important to at least mention the vectors and the parasite type! Is there more than one species of malaria? 

6) I am confused with the testing algorithm Did the study only test children? what are the results of the household parents? other members in the household? 

7) The study avoids addressing the confounding factors. having more household members is associated necessarily with poverty. In your argument you mentioned that increasing the household member by a factor of one resulted in increase in the odds of infection. Does this stand when testing only household members that you identify as rich (not poor or poorest) perhaps middle class or upper class? 

Also does any of the children do any paid labor? has this been considered in the analysis?

Minor comments:

The English language requires serious attention. I recommend that the authors hire a professional editor to address language problems. it was sometimes hard to follow the text. 

the article can use better table structure and layout. Also some graphs would ease the understanding of the text.

Author Response

Reviewer1. Comments

Answers

1. The authors did not define their parameters as they start

Fixed

2. The model ignored completely the presence of windows and window nets

The study used secondary data. There are no windows in the data set.  We did not ignore them and in addition it is not necessarily that all house have cement and windows.

3. I am very confused with the age group description of the children, sometimes they are one sum, other times they are two groups, then on third occasions they are three groups, simplify and perhaps consolidate the groups. 

It is fixed. The age of the children is categorical (6-59 months, 5-9years, 10-14 years)

4. -The paper completely avoided any descriptive graphs. there is such heavy text loaded with number and not enough graphs or even maps to describe the background!

- When you have four locations what makes it rural

-We did not find important to use graphs in descriptive statistics. Descriptive statistics can be done in many ways and table is one of them. Furthermore there is a restriction on the size of the document. The tables can convey exactly the same interpretation as graphs and descriptive statistics can do so to put several of these may be redundant.

-We can not make the map of observations because Malaria Indicator Survey did not provide GPS coordinates for this survey. However, when you are in Rwanda rural and urban are clearly separated Furthermore I am a citizen of Rwanda and this is common knowledge to me.

5. the authors completely ignored the biological questions. They state that women are more at risk of infection but never mentioned why!

We did not ignore biological questions. The objective of our study is not to find the causality. The causality can be another study.

6. I am confused with the testing algorithm Did the study only test children? what are the results of the household parents? other members in the household?

The objective of our study is clear. Our study focused only on children aged 6 months to 14 years. In addition, we used secondary data.

7. The study avoids addressing the confounding factors. having more household members is associated necessarily with poverty. In your argument you mentioned that increasing the household member by a factor of one resulted in increase in the odds of infection. Does this stand when testing only household members that you identify as rich (not poor or poorest) perhaps middle class or upper class?

We have included two-way interaction effects between the potential variables but none was significant, that is why we used only main effect.

8. Also does any of the children do any paid labor? has this been considered in the analysis

No information

Reviewer 2 Report

Thanks for asking me to review this manuscript. This paper assessed the prevalence of malaria among children aged 6 months to 14 years old and identified the factors associated with malaria in this age group. However, there is a clear objective that addresses the two purposes of testable research questions. The approach is appropriate for the objective but minor issues and analyses were performed sufficiently. 

I have identified the following areas with minor issues:

  1. This manuscript will benefit from manuscript proofreading
  2. Page 2, line 71-72: What do you mean by ”To the best of our knowledge, no other study has been carried out with these objectives”. Do you mean no research has ever been done with your stated objectives in Rwanda or all over the world? There are several papers of these objectives in Africa, even in sub-Saharan African.
  3. Page 4, lines 131 – 134: This information should be summarized in a table or plot to back up the interpretation of your results.
  4. You stated that 8209 children were considered but in Table, the total number is 8210 under the province variable
  5. Table 1 is poorly reported. It is either you report the number of children with malaria and its percentage or report both the malaria-infected and non-malaria infected children. You need to report the number of infected in each province to calculate the infected percentage. Each of the number reported is for all the children in the province not infected children. You must re-analysis this descriptive statistic of children's malaria prevalence.
  6. Under the variable ‘has electricity’, the total is 7709 instead of 8209.
  7. Did you use the p-value to adjust for potential confounding variables before using logistic regression? What is the importance of the p-value in Table 1?
  8. Page 4, line 136 – 138: I can not find where these values are reported in your Tables. Residential is not included in your table, why?
  9. Same as in page 4, line 140 – 141: this is not summarized in any of your tables. However, the age in your table1 is continuous while in your text, you reported a categorical age value.
  10. I have seen many of the variables interpreted in your text but not reported in your table. I suggest that the author should only report the main characteristics/information that determines the prevalence of malaria among children. Author(s) is/are reporting unimportant variables.
  11. Very poor interpretation of Tables 2 and 3.
  12. Move both tables 2 and 3 to supplementary files. Both Tables should be interpreted adequately.
  13. All your Tables are poorly labeled and formatted.
  14. Poor interpretation of your multiple regression. Specify the age that is significant. The author (s) just reported that the age of the child is significant. Same in the province and many more.
  15. Page 8, line 250 – 251: how did you get the results of the prevalence of malaria among children aged from six months to 14 years old to be 14.0%?

The current discussion and conclusion of the results are reasonable and allow confidence in the results

Author Response

Reviewer2: Comments

Answers

1. This manuscript will benefit from manuscript proofreading[

Fixed

2. Page 2, line 71-72: What do you mean by ”To the best of our knowledge, no other study has been carried out with these objectives”. Do you mean no research has ever been done with your stated objectives in Rwanda or all over the world? There are several papers of these objectives in Africa, even in sub-Saharan African

Fixed.  As “To the best of our knowledge, no other study has been carried out “in Rwanda “ with these objectives ”

3. Page 4, lines 131 – 134: This information should be summarized in a table or plot to back up the interpretation of your results

I have included Table1

4. You stated that 8209 children were considered but in Table, the total number is 8210 under the province variable

Fixed

5. Table 1 is poorly reported. It is either you report the number of children with malaria and its percentage or report both the malaria-infected and non-malaria infected children. You need to report the number of infected in each province to calculate the infected percentage

Reported in Table 2 as requested

6. Under the variable ‘has electricity’, the total is 7709 instead of 8209

Fixed

7. Did you use the p-value to adjust for potential confounding variables before using logistic regression? What is the importance of the p-value in Table 1?[

Yes, table2

8. -Page 4, line 136 – 138: I can not find where these values are reported in your Tables.

- Residential is not included in your table, why

-Fixed in Table 1 and Table2

-Included in Table1 and Table2

9. Same as in page 4, line 140 – 141: this is not summarized in any of your tables. However, the age in your table1 is continuous while in your text, you reported a categorical age value

-Age of household head is continuous

 -Age of the child is categorical (table1 and 2)

10. I have seen many of the variables interpreted in your text but not reported in your table. I suggest that the author should only report the main characteristics/information that determines the prevalence of malaria among children. Author(s) is/are reporting unimportant variables.

fixed

11. Very poor interpretation of Tables 2 and 3.

This is now done

12 Move both tables 2 and 3 to supplementary files. Both Tables should be interpreted adequately.

-We have put in the greater interpretation here.

13. All your Tables are poorly labeled and formatted

This has now been corrected as the paper was sent to a professional editor.

14. Poor interpretation of your multiple regression. Specify the age that is significant. The author (s) just reported that the age of the child is significant. Same in the province and many more

After the sentence it was clearly interpreted line “241-245” and line 230 for province. It was only reported the significant age group and province

15. Page 8, line 250 – 251: how did you get the results of the prevalence of malaria among children aged from six months to 14 years old to be 14.0%?

Fixed “table 1”

Reviewer 3 Report

This manuscript looks at risk factors for malaria infection in children aged 6 months to 14 years in Rwanda. The authors consider a large dataset collected in 2017 for malaria antigens and a broad set of risk factors.

The manuscript is scientifically sound, the authors present and discuss their findings clearly, by placing them in the broader malaria literature.

The introduction is overall well written, however it may need some improvement before publication (see below). The methods are appropriate for the type of data collected and analysed. It may be that a part of methodology which explains how Table 1 has been achieved is missing, if this is the case, the authors need to add text to match methods with results (see comments below). Discussion and conclusions are interesting and match aims and results of the study.

Detailed comments

42: change where with “as” or rephrase.

52: Anopheles with capital letter

54: It is not clear what is the message that should be associated with the governmental policy action “mutuelle de santè”. The authors define it “tremendous” but then give percentages which seem to show that the incidence of Malaria among among women of child-bearing age increased to 5% from 2015 to 2017. Please explain plainly what was the outcome of the policies on malaria incidence (and perhaps curb the “propaganda” style?).

79: Not clear what “clusters with probability proportional to sample size” mean here. Please explain what probability is proportional to sample size and how it was used for the sampling.

70: Not clear what a “systematic sampling technique” is in this context. Please explain.

101: Please add references which justify what you state in this sentence (“The justification as to the choice of these variables from a rigorous theoretical framework was that these variables have been found to be statistically significant in existing literature”).

122: The formula makes it look like the regression parameter betas were measured only for the cluster (k) level? I have the feeling that the authors use the k-index for the beta as an indication of k-dimensional and not as k as index for the cluster. Change to another letter and match betas with the appropriate level of measurement (household, cluster, etc)

129: Please change to “The alpha level was 0.05” or “The threshold for significance for type I error rate was set to 5%”.

Table 1: I am not sure how the authors tested the differences in incidence among groups reported in Table 1. In the methods authors explain that they used a hierarchical (survey) logistic regression to test the significance of the relationships between malaria incidence and a series of co-variates (risk factors). I do not see any reference to an ANOVA or similar test which seems to have been used in Table 1. Please explain and match the methods with results.

196: In the table you report 8.5% as well as a different p-value.

276: Check the spelling after 2017, also some % are missing

301: remove complex from “complex logistic regression”

Author Response

Reviewer 3: Comments

Answer

1. 42: change where with “as” or rephrase

Changed as suggested

2. 52: Anopheles with capital letter

Fixed

3. It is not clear what is the message that should be associated with the governmental policy action “mutuelle de santè”. The authors define it “tremendous” but then give percentages which seem to show that the incidence of Malaria among among women of child-bearing age increased to 5% from 2015 to 2017. Please explain plainly what was the outcome of the policies on malaria incidence (and perhaps curb the “propaganda” style?).

I have excluded the sentence. But for the prevalence, the references were provided. Explaining the policies is not the objective of this study.

4. Not clear what “clusters with probability proportional to sample size” mean here. Please explain what probability is proportional to sample size and how it was used for the sampling

Cluster is replaced by Primary sampling unit, and probability proportional to sample size is replaced by” probability proportional to number of households in the village

5. 70: Not clear what a “systematic sampling technique” is in this context. Please explain

fixed

6. : Please add references which justify what you state in this sentence (“The justification as to the choice of these variables from a rigorous theoretical framework was that these variables have been found to be statistically significant in existing literature”).

added

7. 122: The formula makes it look like the regression parameter betas were measured only for the cluster (k) level? I have the feeling that the authors use the k-index for the beta as an indication of k-dimensional and not as k as index for the cluster. Change to another letter and match betas with the appropriate level of measurement (household, cluster, etc

fixed, dropped k on beta

8. Please change to “The alpha level was 0.05” or “The threshold for significance for type I error rate was set to 5%”.

Fixed.  Line “150-151”

9. Table 1: I am not sure how the authors tested the differences in incidence among groups reported in Table 1. In the methods authors explain that they used a hierarchical (survey) logistic regression to test the significance of the relationships between malaria incidence and a series of co-variates (risk factors). I do not see any reference to an ANOVA or similar test which seems to have been used in Table 1.

we do not have to use the ANOVA because we are not testing the differences in the mean or average number of incidences among the groups.  After revision Table1 became  table 2 where I have tested the association between RDT malaria and potential variables.

10. 196: In the table you report 8.5% as well as a different p-value

OR=1.088 this the resulted from survey logistic whereas  8.5% is the results from cross-tabulation

11. 276: Check the spelling after 2017, also some % are missing

Fixed

12.  301: remove complex from “complex logistic regression

 Complex was replaced by “Survey”